# Classroom Movement Breaks and Physically Active Learning Are Feasible, Reduce Sedentary Behaviour and Fatigue, and May Increase Focus in University Students: A Systematic Review and Meta-Analysis

**DOI:** 10.3390/ijerph19137775

**Published:** 2022-06-24

**Authors:** Julia Lynch, Gráinne O’Donoghue, Casey L. Peiris

**Affiliations:** 1School of Allied Health, Human Services and Sport, Physiotherapy, La Trobe University, Melbourne, VIC 3086, Australia; c.peiris@latrobe.edu.au; 2UCD School of Public Health, Physiotherapy and Sports Science, University College Dublin, Belfield, Dublin 4, Ireland; grainne.odonoghue@ucd.ie

**Keywords:** exercise, mental fatigue, universities, tertiary, academic performance, college, attention, cognition, cognitive functions

## Abstract

*Background:* University students are mostly sedentary in tertiary education settings which may be detrimental to their health and learning. This review aimed to examine the feasibility and efficacy of classroom movement breaks (CMB) and physically active learning (PAL) on physical and cognitive outcomes in university students in the tertiary setting. *Methods:* Five electronic databases (MEDLINE, CINAHL, Embase, PsychINFO, and PubMed) were searched for articles published up until November 2021. Manual searching of reference lists and citation tracking were also completed. Two reviewers independently applied inclusion and exclusion criteria and completed quality assessment. Articles were included if they evaluated CMB or PAL interventions delivered to university students in a tertiary setting. *Results:* Of the 1691 articles identified, 14 studies with 5997 participants met the inclusion criteria. Average study quality scores were poor for both CMB and PAL studies. CMBs and PAL are feasible in the tertiary setting and increase physical activity, reduce sedentary behaviour, increase wellbeing, and reduce fatigue in university students. In addition, CMBs increased student focus and attention in class and PAL had no detrimental effect on academic performance. *Conclusions:* University educators should feel confident in introducing CMB and/or PAL interventions into their classes to improve student health and wellbeing.

## 1. Introduction

People of all age groups are spending large amounts of time sedentary [1,2,3] and are not achieving physical activity recommendations [4,5,6,7]. Sedentary behaviour is any behaviour whilst awake that utilises ≤1.5 metabolic equivalents (METs) of energy expenditure in a sitting, reclining, or lying position [8], a common behaviour seen in university students in lecture theatres and classrooms and while studying [1]. In addition, there is an observed decline in physical activity in early adulthood [9] which, unlike other behaviours such as smoking and binge drinking of alcohol, is a behaviour that does not resolve [10]. High levels of sedentary behaviour are associated with poor cardiometabolic and mental health in children and adolescents [11] and several chronic diseases in adults including diabetes and cardiovascular disease [4,12]. Conversely, physical activity is associated with increased cardiometabolic health, improved cognitive and academic outcomes, and decreased risk of depression in children and adolescents [11]. In adults it is associated with decreased all-cause mortality [4], improved mental health [13,14], and cognitive function [15]. With increased sedentary behaviour and reduced physical activity within all age groups across the world resulting in poorer physical and cognitive health outcomes, there is a need to determine strategies to facilitate change.

Compared to university students who are more active, those who are less active have an increased chance of obesity [16,17,18] and its associated metabolic risk factors [19,20]. This period of early adulthood can be a critical time for the development of obesity as an increased body mass index (BMI) during this time is associated with higher morbidity, premature mortality, and chronic obesity [21]. As well as the health benefits of physical activity, observational studies have shown increased physical activity is linked to improved cognitive function including executive function in university students [22,23,24], improved mood, lowered stress [25], and improved working memory capacity [26]. In contrast, sedentary behaviour related to uninterrupted sitting has been shown to increase discomfort and sleepiness [27] in this population. Therefore, university students are an important cohort to target with interventions to reduce sedentary behaviour and increase physical activity to improve their current and future health outcomes.

Classroom movement breaks (CMB) (a brief exercise break designed to be time efficient and feasibly implemented in the classroom) [28] and physically active learning (PAL) (combination of physical activity with academic content) [29] are two potential strategies that could be used to reduce sedentary behaviour and increase physical activity in university students, and they may have additional cognitive benefits. Previous systematic reviews found that CMB and PAL interventions are feasible [30,31] and increase physical activity in preschool [29], primary [29,32,33] and secondary [33] students but have limited effect on academic performance [29,32,33]. Published reviews have also synthesised data on the use of active workstations [34,35] (a form of PAL) and physical activity breaks [36] (a form of CMB) in the adult working population and found they were feasible [37], increased physical activity, resulted in positive physiological changes [34,36] and were not detrimental to productivity [34,35]. To our knowledge, there are no reviews of CMB or PAL in university students. Encouragingly lab-based studies of university students have found that low to moderate intensity movement breaks improve cognition [38,39,40], learning and memory [39,41], and executive function [38], and cycling while studying reduced sympathetic reactivity to stress which may be important for learning [42]. However, there is limited evidence on feasibility and effectiveness in tertiary education settings. Therefore, this systematic review seeks to determine whether PAL and CMB are feasible and effective for university students in the tertiary setting.

## 2. Methods

### 2.1. Registration

The systematic review was registered prospectively with PROSPERO, the international register of systematic reviews, in February 2021 (registration number: CRD42021230524). The Preferred Reporting Items for Systematic Reviews and Meta-Analyses (PRISMA) was adhered to when conducting and reporting this review.

### 2.2. Search and Study Selection

A keyword search of the literature was conducted from the earliest date available until 22 November 2021 within the following databases: MEDLINE, CINAHL, Embase (OVID), and PsychINFO. The search was then amended for PubMed (Appendix A). These five databases were chosen as they are all major search platforms that index research related to education, physical, and cognitive health. The search strategy included key words and their synonyms in two categories: ‘intervention’ (e.g., movement break, physically active learning) and ‘population’ (e.g., tertiary, college, university). Synonyms were combined with the ‘OR’ operator and categories were combined with the ‘AND’ operator. No filters were applied. Manual searching of the reference lists of included studies and citation tracking of included studies were also completed via Google Scholar to identify any additional studies. For inclusion in the review, studies had to involve participants who were university/tertiary students at any level of study. The intervention had to be a movement break or physically active learning within a tertiary, university, college, or higher education institutional setting (i.e., classroom, library, or study area). Studies had to be peer reviewed papers published in English. Studies were excluded if they were conducted in university-aged students in a laboratory setting as we wanted to evaluate the real-world feasibility and effect of CMBs and/or PAL. Studies had to assess outcomes related to feasibility, physical outcomes (physical activity, upright time), or cognitive outcomes (concentration, academic results).

### 2.3. Study Selection

Search results were managed in Covidence, an online screening and data extraction tool for systematic review management [43]. Two reviewers (JL and CP) independently applied the inclusion and exclusion criteria to the titles and abstracts of all identified studies to determine which needed to be sourced in full text. Any disagreements were discussed until consensus was reached. If consensus could not be reached, a third reviewer was consulted (GO’D). When articles could not be confidently excluded based on title and abstract, full text articles were obtained, and the same process was applied for selection of articles based on the full text. The agreement between the two reviewers was calculated using the Cohen’s Kappa coefficient (κ) statistic with 0.21 to 0.40 considered fair agreement, 0.41 to 0.60 considered moderate, 0.61 to 0.80 considered substantial, and 0.81 to 0.99 considered almost perfect agreement [44].

### 2.4. Data Extraction

A data extraction form was developed and applied by one reviewer (JL) and checked for accuracy by a second reviewer (CP). Data were extracted on study location, sample size, participant characteristics (age, gender, body mass index, physical activity, academic performance, ethnicity), intervention details (type of movement, duration/intensity, location and supervision of intervention), cognitive outcome measures (e.g., academic results, enjoyment, focus), physical outcome measures (e.g., physical activity, sedentary behaviour), and results. The authors of one study [45] were contacted to obtain missing information.

### 2.5. Study Quality and Risk of Bias

Following PRISMA guidelines, all trials were critically appraised for methodological quality and risk of bias by two independent reviewers (JL and GO’D) using the Downs and Black checklist, a quality assessment tool to assess the methodological quality of randomised and non-randomised trials [46]. The checklist consists of 27 items across five sections and provides an overall rating for study quality and a numeric score out of 32 [46]. The five sections assess reporting (9 items), external validity (3 items), bias (7 items), confounding (6 items), and power (1 item). The scoring of item 27, which refers to the power of the study, was modified. Instead of rating according to an available range of study powers, we rated whether the study did or did not perform a power calculation as per previous reviews [33,47,48]. Accordingly, the maximum score for item 27 was 1 (a power analysis was conducted) instead of 5 and thus the highest possible score for studies in this review was 28 (instead of 32). The modified Downs and Black score ranges were given corresponding quality levels as previously reported [47,48]: excellent (26–28); good (20–25); fair (15–19); and poor (≤14). Any discrepancies between reviewers were resolved through discussion until consensus was reached, if a consensus could not be reached, a third reviewer (CP) was consulted. The agreement between the two reviewers (prior to discussion) was calculated using the Cohen’s Kappa coefficient (κ) statistic. The checklist has high internal consistency, good face and criterion validity, and good re-test and inter-rater reliability [46].

### 2.6. Data Analysis

Where possible, meta-analyses of randomised controlled trial data were planned. Where studies had sufficient homogeneity in terms of interventions and outcomes, post-intervention data were pooled using random effects models and inverse variance methods to calculate mean differences and 95% confidence intervals using RevMan 5 [49]. Where meta-analyses were not possible, a narrative synthesis was completed. Data were grouped based on intervention type (either PAL or CMB) and outcome type (feasibility, physical or cognitive). Feasibility outcomes were grouped according to Bowen’s Framework for feasibility studies [50] which describes eight focus areas that may be addressed by feasibility studies: demand, acceptability, implementation, practicality, adaptation, integration, expansion, and limited efficacy testing. All studies were assessed against these eight focus areas and common focus areas were described. Physical outcomes of interest were time spent sedentary, upright, or active and cognitive outcomes of interest included academic performance, concentration, focus, and attention.

## 3. Results

The initial database search identified 1691 records. After removing duplicates (*n* = 148), 1543 records were screened by title and abstract. Agreement between reviewers was almost perfect (k = 0.870, 95% CI 0.757 to 0.983) when excluding 1527 articles. Sixteen full text articles were then assessed against the eligibility criteria with thirteen studies meeting the inclusion criteria. One further record was found through citation tracking and manual searching of reference lists of included studies. Figure 1 shows the reason for rejection at the full text article stage. Missing information was obtained from authors of one study and was included in this review [45].

### 3.1. Study Characteristics

Of the fourteen studies included, eight featured PAL interventions and six utilised CMB. Studies varied in design and were predominantly feasibility studies (*n* = 7) and randomised controlled trials (*n* = 3), with one each of a non-randomised controlled trial, cross-over intervention, single group intervention control and observational (see Table 1 and Table 2). Of the feasibility studies, five utilised surveys [51,52,53,54,55] and two mixed methods [56,57].

The eight PAL studies included 4800 participants who were aged between 18 and 28 years. Five were conducted in general study areas (e.g., library) [52,53,56,59,60] and three in the classroom [45,51,58]. Most included a variety of students with one being conducted in the sports department [51] and one with exercise physiology students [45]. Most of the PAL studies (88%) were published in the last four years with the first published in 2014. Five studies were conducted in the USA [45,52,56,58,60] and one each in Canada [53], France [51], and Germany [59].

The six CMB studies included 1197 participants aged between 17 and 25 years mostly enrolled in sports or health science courses [55,57,61,62]. All CMB studies were published within the last four years. Two studies were conducted in Austria [61,62] and one each in Australia [57], Republic of Ireland [55], Germany [63], and the USA [54].

### 3.2. Quality Assessment

Average study quality scores were poor for both PAL (mean 12 out of 28, range 6 to 18) and CMB studies (mean 11 out of 28, range 6 to 21). Two CMB studies were considered good quality [57,62] and three PAL studies were of fair quality [45,58,60]. All but one CMB study [54] scored well for reporting and all studies clearly described the interventions and main findings of the study. No studies calculated power and most studies scored poorly on external validity with only three PAL studies [45,51,58] and four CMB studies [55,57,62,63] gaining a score for this item. Whether the participants were representative of all university students was unclear, and there was no blinding of participants or assessors in any study (Table 3). Agreement between reviewers on the Downs and Black quality scores for all included studies was almost perfect (k = 0.931, 95% CI 0.894 to 0.968).

### 3.3. Intervention Design and Delivery

Of the eight PAL studies, four were conducted in the library setting [52,53,56,60], three in a classroom setting [45,51,58], and one in a common study area [59] (Table 1). Three of the PAL studies used a portable peddle machine or static cycling desk [45,51,52], two used a standing desk or sit to stand desk [58,59], and three included both cycling and standing options [51,53,56] as part of their intervention. Most PAL studies were unsupervised with participation self-reported by the students [52,53,56,59,60]. Three PAL studies involved some supervision by a researcher/classroom teacher [45,51,58]. In most of the PAL studies, duration and intensity were self-determined by the participant [51,52,53,56,58,59]. Two studies specified that they aimed to simulate low intensity physical activity [45,60] (Table 1).

All CMB studies were set in a classroom or lecture theatre and comprised 4–10 min of movement without equipment (e.g., walking, running, aerobic exercises) (Table 2). Classroom movement breaks were conducted after 20 [54,57], 45 [61,62,63], or 105 [55] minutes of sedentary class time. All CMB interventions were supervised by academic [54,55,57,63] and/or research staff [53,62,63] with the intensity self-determined by the participants (Table 2).

### 3.4. Feasibility

Half of the included studies assessed feasibility of PAL or CMB interventions in the university setting by evaluating demand, acceptability, practicality, and integration [50].

*Demand*, in terms of active workstation or equipment use and preference, was evaluated in four PAL studies [51,52,53,56]. In the library setting, portable pedal exercise machines were used for an average of 51 [53] to 96 [52] minutes per day and standing desks were used for an average of 69 min per day [53]. Maeda et al. [52] observed pedal machines were used 15% of the time and Clement et al. [56] reported that 25% of students used the active learning space one to two times a week. In one study, standing desks were the preferred option for PAL [56], but in another, cycling desks and balance balls were preferred [51]. Bastien Tardiff et al. [53] found significant differences dependent on gender, with women participants preferring a cycling desk and men participants preferring a standing desk.

The *acceptability* of PAL and CMB was addressed by seven PAL studies [45,51,52,53,56,58,60] and four CMB studies [54,55,57,63] in terms of satisfaction with intervention, intention to continue to use intervention, and perceived appropriateness of intervention. Satisfaction with PAL was explored in three studies [52,53,56] with most participants reporting they were in favour of active workstations [53], that active learning space helped their studying [56], and cycling desks were comfortable to use [52]. Two PAL studies demonstrated acceptability by reporting a lack of distraction and an ability to complete usual tasks while participating in PAL interventions [45,52]. The majority of PAL studies [45,51,52,53,58,60] measured intention to continue the use of PAL by participants. All studies reported that most participants would use PAL if available [45,51,53,58,60] or would prefer PAL to continue to be available in the future [52,56]. In three studies most participants (66% to 73%) reported they were satisfied with the CMB interventions [54,55,57]. Two CMB studies measured desire to continue CMBs with 93% of participants in one study approving the continuation of an active movement break [63] and participants rating their interest in continuing CMBs in other classes as 6.4 out of 10 [54]. CMBs were considered highly appropriate and not disruptive to class productivity in three studies [55,57,63].

Two PAL studies [51,56] and two CMB studies [57,63] assessed *practicality* by considering the positive and negative effects of the application of the intervention and ability to carry out the intervention. Grospretre et al. [51] recorded that half of their lecturers reported that the quality of the lecture was unaffected by the introduction of a PAL and 71% would continue to have active workstations in their classrooms despite 43% finding delivering the lecture more difficult [51]. Clement et al. [56] reported that two participants had raised the need for adequate instructions for active workstation use which could have affected participants ability to participate in the intervention. Both CMB studies [57,63] reported that careful planning and flexibility were required in the timing and duration of breaks to optimise benefits and reduce disruption.

One PAL study [56] and one CMB study [55] assessed *integration* in relation to the current infrastructure of the setting. In addition to the PAL equipment, Clement et al. [56] reported the need for increased electrical outlets within the active learning space. Keating et al. [55] reported that space and seating constraints were limitations for integrating CMBs into classes.

### 3.5. Physical Outcomes

Students in PAL studies spent an average of 115 min per week each on a stationary bike whilst studying [60] and 213 min cycling in class per week [45]. Students with access to sit to stand desks in the classroom setting stood for six to eight minutes per hour compared to students with seated desks who stood for zero to two minutes per hour [58]. In classrooms with multiple active workstation options, 35% of students were active or upright for more than one hour of a two-hour class [51]. Prompts to increase physical activity whilst studying increased standing time from 6% of study time to 11% of study time [59].

Two CMB studies recorded physical activity levels during class. In one study, students took 834 (95%CI 675 to 994) more steps and spent 10 (95% CI 8 to 12) more minutes walking in classes with movement breaks compared to classes without movement breaks [57]. In the other, students ran 1100 to 1500 m in a 10 min movement break compared to those who were not in the movement break group who remained in the lecture theatre [62].

One PAL study [45] and one CMB study [63] reported changes in participants’ behaviour because of the intervention. Joubert et al. [45] reported that the PAL group participants described an increase in daily physical activity (outside of class) and Paulus et al. [63] reported that active break and standing break participants were inspired to break up their sitting more frequently in other settings.

### 3.6. Cognitive Outcomes

Academic performance was assessed in two randomised controlled trials of PAL interventions [45,60]. Considering the meta-analysis of two trials with 138 participants, studying on a stationary bike had no significant effect on final course grades compared to using a standard desk (MD 2.16, 95% CI −0.78 to 5.09, I^2^ 0%) (Figure 2).

One PAL study reported that 92% of participants found that the active learning space contributed positively to their wellbeing [56]. In single-group PAL studies, 20% [51] to 51% [58] of participants reported an increase in attention, and 36% reported an increase in focus [58]. However, there were no significant differences in self-reported focus between intervention and control groups in a randomised controlled trial [60]. Eighteen [51] to 35% [58] of students reported greater engagement in class, 46% [58] to 58% [51] reported less boredom and 36% [51] to 39% reported less distraction and reduced use of their cell phones during class when engaged in PAL [58]. Self-reported mental fatigue was shown to decrease by 34% [51] to 43% [58] and anxiety levels were reported to be reduced by 20% [58] to 40% [51] of participants in two PAL studies.

Academic performance was not assessed in any CMB study. Compared to no break, self-reported vigour and concentration were increased, and fatigue decreased significantly (*p* < 0.01) immediately following an active CMB with improvements sustained for 20 min following the break in a single-group intervention–control study [61]. Paulus et al. [63] found 80% of participants who partook in a CMB felt it improved their wellbeing in terms of balance, vigour, and motivation and 91% reported that it improved their concentration. Students reported significantly higher alertness, concentration, and enjoyment when participating in two-hour classes with three CMBs compared to classes without movement breaks (*p* < 0.01) in another single-group intervention–control study [57]. Compared to a sedentary control group, students who went for a 10 min run halfway through a two-hour lecture had significantly higher objectively measured visual attention scores (*p* = 0.003) and higher perceived attention and arousal (*p* < 0.01) in a randomised controlled trial [62].

## 4. Discussion

The results of this systematic review of 14 trials with 5997 participants provide evidence that CMB and PAL are feasible for use amongst university students in tertiary education settings. Students and tutors found both interventions acceptable with students keen to continue them as they were not distracting and fitted into class time, consistent with findings from systematic reviews in primary school children [30], adolescents [31], and adults [35]. Therefore, university educators should feel confident to consider the incorporation of CMB and/or PAL into their classrooms if they have the space and infrastructure to do so. Furthermore, there is preliminary evidence that CMB and PAL reduce sedentary behaviour and increase physical activity in university students. Students who completed PAL self-reported improved wellbeing, no change or improved focus and attention, and decreased fatigue. There were no differences in academic performance between students who did or did not participate in PAL, while academic performance was not evaluated in CMB studies. The results indicate that PAL and CMBs have similar effects on university students as they do in primary school children, adolescents in secondary school, and adults in the workplace. Lab-based studies on university students are also in agreement with the results of the current review finding that PAL improved student wellbeing [42] and may improve executive function [64].

By replacing previous sedentary behaviour during class/study time with PAL or CMB, university students’ physical activity during class/study time is increased and sedentary behaviour is decreased as seen in school-aged students [33]. However, the amount of change may be insufficient to affect overall health unless it has carry-over effect on behaviour. In the studies included in this review, students completed an average of 10 min of physical activity in CMB studies and stood for 8 to 60 min in class in PAL studies. This may have had minimal effect on weekly physical activity levels considering guidelines recommended at least 150–300 min of moderate intensity physical activity per week [4]. In one study, students completed an average of 213 min per week of low intensity cycling which may have been sufficient to influence health [45]. There did appear to be some carry-over effect of the intervention to out of class behaviour in two studies where students reported increasing their overall physical activity, being more aware of their physical activity habits [45] and reducing their sedentary behaviour by regularly interrupting their sitting with movement [63]. However, to positively influence health, long-term behaviour change and health promotion strategies may need to be considered in conjunction with PAL or CMB interventions. A recent systematic review of sedentary behaviour in university students highlighted the need to include the awareness of negative health outcomes, productivity concerns, training in behavioural self-regulation, habit formation techniques, and social acceptability of breaking up sedentary behaviour to increase the likelihood of sustained behaviour change [65]. Therefore, combining PAL and CMB interventions with education and behaviour change strategies may have the potential to have a positive influence on health outcomes.

The observed cognitive benefits of PAL and CMBs in relation to focus, attention, fatigue, and wellbeing may be partially explained by the influence of physical activity on a variety of physiological mechanisms as breaking up prolonged sitting with intermittent standing or low to moderate physical activity results in metabolic, endocrine, and vascular changes [66]. This may improve cognitive function by decreasing postprandial hyperglycaemia, insulin resistance, inflammatory markers, and by improving hormonal regulation and blood flow to cortical and peripheral arteries [66]. Despite the observed cognitive benefits and the potential physiological rational for these benefits, CMBs did not result in improved academic performance in the studies included in this review. This may be because only two studies measured academic performance, but it may also partially be due to the type and timing of the movement breaks utilised. The timing, intensity, and duration of the physical activity completed during a movement break may have an important role in the amount of physiological and cognitive change. For example, in response to incremental exercise there is an increase in cerebral blood flow until approximately 60% of maximal intensity which then plateaus before declining back towards resting levels when approaching maximal intensity [67]. Therefore, intensity needs to be sufficient to raise cerebral blood flow, which some exercises such as calf raises may not be [68], but not be too intense or too long to elicit prolonged hypoxia which is associated with declines in cerebral blood flow [67]. In addition to intensity, timing of movement breaks is an important consideration. In a study of office workers, drops in cerebrovascular blood flow due to sedentary behaviour were reversed with 2 min of light intensity walking at self-selected, habitual walking speed every 30 min but not with an 8 min walk every 2 h at the same walking speed [69]. For glucose regulation, 3 min of low to moderate intensity physical activity after 30 min of sedentary time is considered the minimum dose to elicit changes in glucose regulation [70]. Although type and timing of CMB and PAL varied, there is sufficient evidence to suggest that CMBs and PAL have a positive effect on cognition and whilst there is no significant change in academic performance there is also no significant decline, giving confidence to academic teaching staff that CMBs and PAL are safe to implement. Future research should consider measuring academic success and comparing timing, intensity, and duration of PAL and CMB in terms of cognitive and health benefits. In a learning environment which can be emotionally challenging, further research investigating the role CMBs and PAL in emotional regulation may also be warranted to enable a multifaceted understanding of the effects of CMBs and PAL in a tertiary setting.

### Limitations

To our knowledge, this is the first review to evaluate PAL and CMB in university students. It was registered prospectively and conducted according to the PRISMA guidelines. A limitation was that the review only included three randomised controlled trials and the average quality score of the studies included in the review was low. However, low-quality studies had similar results to high-quality studies. Another limitation is that the majority of included CMB studies were performed in health sciences or sports science university students who potentially already have a higher baseline physical activity level and may be more agreeable to PAL and CMB interventions. Only two PAL studies evaluated academic performance; therefore, the results should be interpreted with caution.

## 5. Conclusions

This systematic review of 14 trials with almost 6000 university students found that CMBs and PAL are feasible in the tertiary education setting and increase physical activity, reduce sedentary behaviour, increase wellbeing, and reduce fatigue in university students. In addition, CMBs increased student focus and attention in class and PAL had no detrimental effect on academic performance. University educators should feel confident in introducing either CMB and/or PAL interventions into their classes to improve student health and wellbeing.

## Figures and Tables

**Figure 1 ijerph-19-07775-f001:**
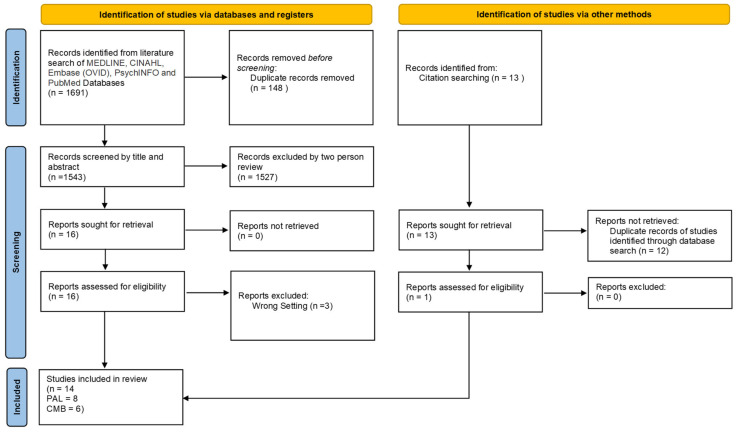
PRISMA flow chart illustrating study inclusions through stages of the systematic review. Abbreviations: PAL, physically active learning; CMB, classroom movement break; PRISMA, Preferred Reporting Items for Systematic Reviews and Meta-Analyses.

**Figure 2 ijerph-19-07775-f002:**
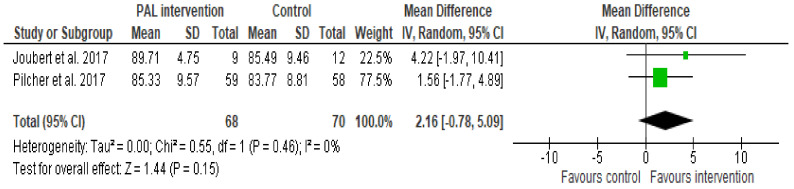
Meta-analysis of two trials of studying using stationary bikes vs. standard desk effect on final course grades [45,60].

**Table 1 ijerph-19-07775-t001:** PAL study characteristics.

Study, Design, Location	Participants	Baseline Characteristics	Intervention	Feasibility Outcomes	Efficacy Outcomes
Bastien Tardif et al. [53], feasibility study, Canada	*n* = 99 Mean age: 28 years Gender: 43 M, 51 F, 5 NB	BMI: 23.6 63% meeting PA guidelines Sedentary Time: 7.7 h/day Physical Activity: 4.1 h/day	Experimental: Choice of portable pedal exercise machine or standing desk When: Participant determined Where: Library Duration: 5 months Supervision: Participant self-supervised Control: Conventional sitting desk	Qualitative questions on reason for choosing workstation type and barriers to use Questionnaire on desirability, intention to use and possibility for future use of active workstation	Cognitive outcomes None Physical outcomes Duration of use of active workstation
Clement et al. [56], mixed-methods feasibility study, USA	*n* = 138 Age: NR Gender: NR		Experimental: Choice of standing desk, stationary bike, workstations, treadmill desks, or balance-ball chairs When: Participant determined Where: Library Duration: 5 months Supervision: Researcher and participant self- supervised Control: Standard desk and chair	Survey of usage and perceptions of active learning space	Cognitive outcomes Survey of perceived wellbeing Physical outcomes Ethnographic observations of equipment use
Grospretre et al. [51], feasibility study, France	*n* = 663 Mean age: 19 years Gender: 417 M, 246 F	BMI 21.6 Sedentary Time: 82% ≥4 h/day 25% ≥7 h/day PA: 40% >10 h/week, 50% 5–10 h/week	Experimental: Choice of Standing desks, swiss ball, upright cycling desk, or seated pedal/stepper board When: During class, participant determined intensity Where: Classroom Duration: 6 months Supervision: Classroom teacher and participant self-supervised Control: Standard desk and chair	Student and lecturer survey on feasibility and acceptability	Cognitive Outcomes Student and lecturer survey of perceived effect on attention, boredom, stress, participation, distraction, and comprehension Physical Outcomes Estimation of duration of active workstation use
Jerome et al. [58], cross-over intervention, USA	*n* = 496 *n* survey = 143 Mean age: 20 years Gender: 37 M, 106 F	BMI: 23.3 45% meeting PA guidelines Ethnicity: 86% white	Experimental: Height adjustable sit-stand desks and stools, point of decision prompt When: Participant determined Where: Classroom Duration: 12 weeks Supervision: Participant self-supervised Control: Traditional seated desk with armrests	Survey of perceived acceptability	Cognitive Outcomes Survey of perceived changes in health and engagement Physical Outcomes Time spent standing Sit–stand transitions/hour
Joubert et al. [45], randomised controlled trial, USA	*n* = 9 Age: 19–24 years Gender: 7 M, 17 F	PA score ^^^ 346.7 min/week GPA 3.3	Experimental: Stationary cycle desks When: During class at intensity RPE 2/10 for 50 min Where: Classroom Duration: 13 weeks Supervision: Classroom teacher and participant self-supervised Control: Standard tables and chairs	Post-intervention survey on perceptions	Cognitive Outcomes Academic performance class test scores and overall course grade Physical Outcomes Time spent cycling in class, distance, and RPE
Maeda et al. [52], feasibility study, USA	*n* = 527 Mean age: 26 years Gender: NR		Experimental: Portable pedal machines at desks. Prompts encouraging pedal machine use When: Participant determined Where: Library Duration: 11 weeks Supervision: Participant self-supervision Control: None	Survey assessing attitudes towards intervention	Cognitive Outcomes None Physical Outcomes Mean pedal time per day Pedal machine use
Mnich et al. [59], observational study, Germany	*n* = 2809 Age: NR Gender:1882 M, 927 F		Experimental: Sit–stand desks and decisional cues When: Participant determined Where: Study area Duration: 3 weeks Supervision: Researchers Control: Baseline (no decisional cues)	None	Cognitive Outcomes None Physical Outcomes Observed sitting, standing, and active time
Pilcher et al. [60], randomised controlled trial, USA	*n* = 59 Mean age: 18 years Gender: 42 M, 75 F		Experimental: Stationary bike with desktop When: 2 h weekly at slow pace Where: Library Duration: 10 weeks Supervision: Participants self-supervision Control: Standard desk and chair	None	Cognitive Outcomes Academic performance Survey of motivation, morale, engagement, focus, commitment, and perceived effectiveness Physical Outcomes Time spent on stationary bike Physical exertion Sleep quality and quantity

Experimental = experimental group, Control = control group, M = male, F = female, NB = non-binary, NR = not reported, PA guideline = 150 min of moderate vigorous physical activity per week, ST = sitting time, PA = physical activity, ^^^ PA score = duration (min/week) × intensity (1 = low, 2 = moderate, 3 = high) × frequency (times per week of aerobic PA) + estimated minutes per week of general resistance training, RPE = rate of perceived exertion (on scale 1 = no exertion, 10 = maximal effort).

**Table 2 ijerph-19-07775-t002:** CMB study characteristics.

Study, Design, Location	Participants	Baseline Characteristics	Intervention	Feasibility	Efficacy
Blasche et al. [61], single-group intervention–control study, Austria	*n* = 66 Mean age: 23 years Gender: 13 M, 53 F		Experimental: 6 min unstructured break or exercise break or relaxation break When: After 45 min of 2 h lecture Where: Classroom lecture setting Duration: 4 weeks Supervision: Supervised by research assistants Control: No break	None	Cognitive Outcomes Wellbeing questionnaire on fatigue and vigour Physical Outcomes None
Ferrer and Laughlin [54], feasibility study, USA	*n* = 53 Mean age: NR Gender: NR		Experimental: Exercise break of unspecified duration When: Every 15 to 20 min in class Where: Classroom setting Duration: One semester Supervision: Supervised by professors Control: None	Survey on perceptions of exercise breaks	Cognitive Outcomes None Physical Outcomes None
Keating et al. [55], feasibility study, Republic of Ireland	*n* = 106 Age: 17–25+ years Gender: 18 M, 87 F, 1 ND		Experimental: 4 min of simple moderate intensity aerobic exercises When: After 1 h and 45 min of 2 h lecture Where: Classroom/lecture theatre setting Duration: One month Supervision: Researcher Control: None	Questionnaire to students on acceptability, appropriateness, and feasibility	Cognitive Outcomes None Physical Outcomes None
Niedermeier et al. [62], randomised controlled trial, Austria	*n* = 51 Mean age: 22 years Gender: 34 M, 17 F	BMI: 22.7 intervention group, 22.6 control group	Experimental: Running for 10 min at 13–15 Borg RPE intensit yWhen: After 45 min of lecture, for 10 min Where: Outside of classroom Duration: One year Supervision: Test leader and researcher Control: No break	None	Cognitive Outcomes Visual attention with modified trail making test (Zahlen-Verbindungs-Test) Perceived Attention and Affective States Questionnaire Physical Outcomes Distance ran during movement break
Paulus et al. [63], non-randomised controlled trial, Germany	*n* = 836 Mean age: NR Gender: NR	Baseline activity: 96% spend entire 90 min lecture sitting. In total, 2% interrupted with standing breaks, 15% interrupt with stretching exercise and 4% leave lecture due to long sitting	Experimental: 5 min standing break or active break When: After 45 min of class. Where: Classroom or lecture setting Duration: One semester Supervision: By class lecturer, sports student, or participant self-supervised Control: Open break	Survey of student and lecturer opinions of acceptability and practicality of breaks	Cognitive Outcomes Survey of concentration, receptiveness, retention, motivation, and wellbeing Physical Outcomes None
Peiris et al. [57], mixed-methods feasibility study, Australia	*n* = 85 Mean age: 23 Gender: 26 M, 58 F, I NB		Experimental: 5–10 min whole body exercise break When: After every 20 min of a 2 h class Where: Classroom setting and outside of classroom Duration: One semester Supervision: By classroom tutors and participant self-supervised Control: No break	Focus group interviews with students and tutors on feasibility	Cognitive Outcomes Survey on perceived concentration, mental alertness, enjoyment using self-administered 10 cm VAS Physical Outcomes Objectively measured physical activity during class

Experimental = experimental group, Control = control group, M = male, F = female, NB = non-binary, NR = not reported, MET min = energy expenditure metabolic equivalent minutes, VAS = visual analogue scale.

**Table 3 ijerph-19-07775-t003:** Results of the Downs and Black methodological quality assessment ranked by overall quality percentage score.

	Reporting	External Validity	Internal Validity (Bias)	Internal Validity (Confounding)	Power	Total
**Question Numbers**	**1–10**	**11–13**	**14–20**	**21–26**	**27**	
**Maximum Score**	**11**	**3**	**7**	**6**	**1**	**28**
PAL studies						
Bastien Tardif et al. [53]	6	0	3	0	0	9
Clement et al. [56]	4	0	2	0	0	6
Grospretre et al. [51]	5	1	3	0	0	9
Jerome et al. [58]	8	1	4	4	0	17
Joubert et al. [45]	8	1	5	4	0	18
Maeda et al. [52]	6	0	2	0	0	8
Mnich et al. [59]	8	0	4	0	0	12
Pilcher et al. [60]	8	0	4	3	0	15
CMB studies						
Blasche et al. [61]	6	0	4	4	0	14
Ferrer and Laughlin [54]	2	0	3	1	0	6
Keating et al. [55]	6	2	2	0	0	10
Niedermeier et al. [62]	9	3	5	3	0	20
Paulus et al. [63]	6	1	4	2	0	13
Peiris et al. [57]	10	3	5	3	0	21

## Data Availability

Not applicable.

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
