# Peer review of "Classroom Movement Breaks and Physically Active Learning Are Feasible, Reduce Sedentary Behaviour and Fatigue, and May Increase Focus in University Students: A Systematic Review and Meta-Analysis"

_ijerph, 2022, doi:10.3390/ijerph19137775_

Round 1
Reviewer 1 Report
Congratulations to the authors for the work. I wanted to ask you for some clarifications and improvement of some aspects:
- The exact date of the search in the databases must be specified.
- It could have been expanded the number of databases consulted. Could you comment why those were chosen?
- The full search strategy should appear in a review.
- You should specify the filters used in the databases.
Author Response
Thank you for the opportunity to revise and improve our manuscript. Below you will find a point-by-point response to each of the reviewer's comments which we feel has helped to clarify and strengthen our manuscript.
Reviewer 1
1. The exact date of the search in the databases must be specified.
Response: As suggested, we have added this detail to the methods as follows: ‘A keyword search of literature was conducted from the earliest date available until November 22, 2021’.
2. It could have been expanded the number of databases consulted. Could you comment why those were chosen?
Response: We searched 5 major databases which were able to detect most relevant studies for inclusion in our review. To complement our search, we completed reference list scanning and citation tracking of included papers. As evidence that the search was likely adequate, we only identified 1 additional paper through reference list scanning and citation tracking. We have added details to the methods as follows: ‘These five databases were chosen as they are all major search platforms that index research related to education, physical and cognitive health.’
3. The full search strategy should appear in a review.
Response: The full search strategy and altered PubMed search are provided in Supplementary file 1.
4. You should specify the filters used in the databases.
Response: We did not use any filters in our database searches. We have added this detail to the methods section as follows: ‘No filters were applied.’
Reviewer 2 Report
This manuscript entitled “ Classroom movement breaks and physically active learning are feasible, reduce sedentary behaviour and fatigue, and may in-crease focus in university students: a systematic review and meta-analysis” primarily aimed to investigate the effect of CMB and PAL on the physical and cognitive outcomes of university students. The results of this study provide guidance for publilc health and
tertiary education. While it is a very interesting topic. But I think this manuscript has some flaws to fill in before it can be published in a journal. There are several questions should be addressed, which list below. I give a minor revision for this manuscript.
Specific comments
1. In the Abstract part, “Five electronic databases were searched for articles published up until November 2021.” The reviewer suggested that the authors list the specific name of the 5 databases in the abstract part.(Line 15-16)
2. In the Introduction part. “a common behaviour seen in university students in lecture theatres and classrooms.” Please add a reference to support this sentence. (Line 34-35)
3. “and several chronic diseases in adults…” which chronic disease, please provide more detailed information. (Line 39)
4. “University students who do not achieve recommended levels of physical activity” I suggest that the authors add some description of “recommended levels of physical activity” to this paragraph. (Line 46-47)
5. “Classroom movement breaks (CMB) (a brief exercise break designed to be time efficient and feasibly implemented in the classroom) and physically active learning (PAL)…”Abbreviations that exist in articles should be explained when they first appear, such as Classroom movement breaks (CMB) and physically active learning (PAL).
6. In the Methods part. “The systematic review was registered prospectively with PROSPERO” the reviewer suggestd that the authors add a secondary heading to this paragraph. (Line 80-84)
7. In the Results part. Please increase the font size in figure 1 and figure 2. (Line 179)
8. In the Discussion part.“focus and attention and decreased fatigue.” Fatigue is a very wide range, please specify the kind of fatigue. (Line 382-384)
9. “but not with an eight minute walk every two hours at light intensity…” What is light intensity, and how do you define the motion of each intensity, please add a specific description. (Line 434-436)
10. In the Conclusion part. In the opinion of the reviewer, the description in the conclusion part was too verbose, and the reviewer suggests that the authors should abbreviate the section and focus on the main findings of this study. (Line 461-472)
11. some recently studies could be also considered in the discussion, such as
A. Benefits of Two 24-Week Interactive Cognitive–Motor Programs on Body Composition, Lower-Body Strength, and Processing Speed in Community Dwellings at Risk of Falling: A Randomized Controlled Trial. Int. J. Environ. Res. Public Health 2022, 19, 7117. https://doi.org/10.3390/ijerph19127117
A Comparison of Physical Activity and Sedentary Lifestyle of University Employees through ActiGraph and IPAQ-LF. Physical Activity and Health, 6(1), 5–15. DOI: http://doi.org/10.5334/paah.163
Author Response
Thank you for the opportunity to revise and improve our manuscript. Below you will find a point-by-point response to each of the reviewer's comments which we feel has helped to clarify and strengthen our manuscript.
Reviewer 2
- In the Abstract part, “Five electronic databases were searched for articles published up until November 2021.” The reviewer suggested that the authors list the specific name of the 5 databases in the abstract part.(Line 15-16)
Response: As suggested, we have added these details to the abstract: ‘Five electronic databases (MEDLINE, CINAHL, Embase, PsychINFO and PubMed) were searched…’
- In the Introduction part. “a common behaviour seen in university students in lecture theatres and classrooms.” Please add a reference to support this sentence. (Line 34-35)
Response: As suggested, we have added a reference to this statement: Castro, O., Bennie, J., Vergeer, I. et al. How Sedentary Are University Students? A Systematic Review and Meta-Analysis. Prev Sci 21, 332–343 (2020). https://doi.org/10.1007/s11121-020-01093-8. It now reads: ‘Sedentary behaviour is any behaviour whilst awake that utilises <1.5 metabolic equivalents (METs) of energy expenditure in a sitting, reclining, or lying position [8], a common behaviour seen in university students in lecture theatres and classrooms and while studying [1].’
- “and several chronic diseases in adults…” which chronic disease, please provide more detailed information. (Line 39)
Response: As suggested, we have added details relevant to the references that follow: ‘…several chronic diseases in adults including diabetes and cardiovascular disease’.
- “University students who do not achieve recommended levels of physical activity” I suggest that the authors add some description of “recommended levels of physical activity” to this paragraph. (Line 46-47)
Response: Because of the various definitions used in the studies referenced, we have modified this sentence to state: ‘Compared to university students who are more active, those who are less active have an increased chance of obesity [16-18] and its associated metabolic risk factors [19, 20].’
- “Classroom movement breaks (CMB) (a brief exercise break designed to be time efficient and feasibly implemented in the classroom) and physically active learning (PAL)…”Abbreviations that exist in articles should be explained when they first appear, such as Classroom movement breaks (CMB) and physically active learning (PAL).
Response: Please note Classroom movement breaks (CMB) and physically active learning (PAL) are written out in full followed by the abbreviation in the abstract. The first time they are mentioned in the introduction they are also written out in full, and a definition provided as follows: ‘Classroom movement breaks (CMB) (a brief exercise break designed to be time efficient and feasibly implemented in the classroom) [28] and physically active learning (PAL) (combination of physical activity with academic content) [29]’. If the editors would like us to add the definition to the abstract, we will be happy to do so.
- In the Methods part. “The systematic review was registered prospectively with PROSPERO” the reviewer suggested that the authors add a secondary heading to this paragraph. (Line 80-84)
Response: As suggested, the secondary heading ‘2.1 Registration’ has been added and the subsequent sub-heading numbering has been altered.
- In the Results part. Please increase the font size in figure 1 and figure 2. (Line 179)
Response: As suggested, figure 1 has been re-orientated to increase font size and figure 2 has been enlarged.
- In the Discussion part.“focus and attention and decreased fatigue.”Fatigue is a very wide range, please specify the kind of fatigue. (Line 382-384)
Response: The term ‘fatigue’ can refer to cognitive, physical or mental fatigue (among others). The studies included did not specify which type of fatigue they measured: in two studies (Jerome et al. 2017; Grospretre et al. 2021) it was described as self-reported ‘fatigue’ and in one study it was described as ‘sensations of drowsiness and energy depletion… tiredness’ (Blasche et al. 2018). These seem to fit well with the definition of mental fatigue so we have clarified this in the results and discussion:
- ‘Self-reported mental fatigue was shown to decrease by 34% [52] to 43% [59]…’
- ‘Students who participated in CMB self-reported improved wellbeing, focus and attention and decreased mental fatigue’.
- “but not with an eight minute walk every two hours at light intensity…” What is light intensity, and how do you define the motion of each intensity, please add a specific description. (Line 434-436)
Response: We have checked the paper we are referring to here and they have not specified how they defined light intensity except to say that it was light intensity based on participants self-selected habitual walking speed. We have clarified this in the manuscript as follows: ‘In a study of office workers, drops in cerebrovascular blood flow due to sedentary behaviour were reversed with two minutes of light intensity walking at self-selected, habitual walking speed every 30 minutes but not with an eight minute walk every two hours at the same walking speed.’
- In the Conclusion part. In the opinion of the reviewer, the description in the conclusion part was too verbose, and the reviewer suggests that the authors should abbreviate the section and focus on the main findings of this study. (Line 461-472)
Response: As suggested, we have deleted reference to future research in the conclusion and have removed 53 of 131 words to make this section more succinct.
- some recently studies could be also considered in the discussion, such as- Benefits of Two 24-Week Interactive Cognitive–Motor Programs on Body Composition, Lower-Body Strength, and Processing Speed in Community Dwellings at Risk of Falling: A Randomized Controlled Trial. Int. J. Environ. Res. Public Health 2022, 19, 7117. https://doi.org/10.3390/ijerph19127117
A Comparison of Physical Activity and Sedentary Lifestyle of University Employees through ActiGraph and IPAQ-LF. Physical Activity and Health, 6(1), 5–15. DOI: http://doi.org/10.5334/paah.163
Response: Thank you for these suggestions. We read both papers with interest and took note of the cognitive benefits of physical activity noted by Rosado et al. (2022) and the differences in subjectively vs objectively measured physical activity reported by low levels of physical activity reported by Safi et al. (2022). However, due to the populations the studies were conducted in, i.e. older adults (Rosado) and university employees (Safi), we did not find a way of incorporating them into the discussion without distracting from the main message.
Reviewer 3 Report
Thank you for the opportunity and congratulations on your work.
Some minor questions:
Lines 98 to 100: The intervention had to be a movement break, physically active learning or self-directed movement within a Tertiary, University, College, or Higher Education institutional setting (i.e., classroom, library, or study area).
QUESTION: “self-directed movement” has not been referred to before: can authors explain what it means and why just now the reference? Is it a third type in addition to PAL and CMB?
Lines 134 to 135: “…using a modified version of the Downs and Black [46]”
QUESTION: “46” is the original or the modified version? If it is the original, what is the reference of the modified one?
Lines 135 and 136: “; a quality assessment tool to assess the methodological quality of randomised and non-randomised trials [47].
QUESTION: This sentence seems to be missing something…; besides, [47] reference, at the final list, seems to be missing something, also:
- 23 of May 2022]; Available from: https://www.nccmt.ca/knowledge-repositories/search/9.
Lines 170 to 172: “The initial database search identified 1691 records and one further record was found through citation tracking and manual searching of reference lists. After removing duplicates (n=148), 1543 records were screened by title and abstract.”
QUESTION: if You had 1691 records and another from manual searching, You had 1692 record (right?); thus, if you removed 148 duplicates, You should finish having 1544, not 1543…
Lines 415 to 416: “The observed cognitive benefits of PAL and CMBs in relation to focus, attention, fatigue and wellbeing…”
QUESTION: is fatigue exclusively a cognitive feature? How was fatigue evaluated? I presume that students self reported if they had more or less fatigue but did You ask them if it was cognitive or physical fatigue?
Author Response
Thank you for the opportunity to revise and improve our manuscript. Below you will find a point-by-point response to each of the reviewers’ comments which we feel has helped to clarify and strengthen our manuscript.
Reviewer 3
- Lines 98 to 100: The intervention had to be a movement break, physically active learning or self-directed movement within a Tertiary, University, College, or Higher Education institutional setting (i.e., classroom, library, or study area).
QUESTION: “self-directed movement” has not been referred to before: can authors explain what it means and why just now the reference? Is it a third type in addition to PAL and CMB?
Response: Since our interventions of interest were classroom movement breaks and physically active learning we have deleted ‘self-directed movement’ from here.
- Lines 134 to 135: “…using a modified version of the Downs and Black [46]”
QUESTION: “46” is the original or the modified version? If it is the original, what is the reference of the modified one?
Response: This reference refers to the original version. Later on in the paragraph we describe the modification and reference where this has been used in previous reviews as follows: ‘The scoring of item 27, which refers to the power of the study, was modified. Instead of rating according to an available range of study powers, we rated whether the study did or did not perform a power calculation as per previous reviews [33, 48, 49].’
For clarity, we have removed ‘a modified version of’ from the first sentence.
- Lines 135 and 136: “; a quality assessment tool to assess the methodological quality of randomised and non-randomised trials [47].
QUESTION: This sentence seems to be missing something…; besides, [47] reference, at the final list, seems to be missing something, also:
- 23 of May 2022]; Available from: https://www.nccmt.ca/knowledge-repositories/search/9.
Response: We have reviewed this sentence and the reference (which was incorrect). It now reads: ‘Following PRISMA guidelines, all trials were critically appraised for methodological quality and risk of bias by two independent reviewers (JL and GO’D) using the Downs and Black checklist, a quality assessment tool to assess the methodological quality of randomised and non-randomised trials [46].’
- Lines 170 to 172: “The initial database search identified 1691 records and one further record was found through citation tracking and manual searching of reference lists. After removing duplicates (n=148), 1543 records were screened by title and abstract.”
QUESTION: if You had 1691 records and another from manual searching, You had 1692 record (right?); thus, if you removed 148 duplicates, You should finish having 1544, not 1543…
Response: The additional articles was found after the initial screening through reference list scanning and citation tracking of included papers. We have now clarified this in the results as follows: ‘The initial database search identified 1691 records. After removing duplicates (n=148), 1543 records were screened by title and abstract. Agreement between reviewers was almost perfect (k=0.870, 95% CI 0.757 to 0.983) when excluding 1,527 articles. Sixteen full text articles were then assessed against the eligibility criteria with 13 studies meeting inclusion. One further record was found through citation tracking and manual searching of reference lists of included studies’.
- Lines 415 to 416: “The observed cognitive benefits of PAL and CMBs in relation to focus, attention, fatigue and wellbeing…”
QUESTION: is fatigue exclusively a cognitive feature? How was fatigue evaluated? I presume that students self reported if they had more or less fatigue but did You ask them if it was cognitive or physical fatigue?
Response: The term ‘fatigue’ can refer to cognitive, physical or mental fatigue (among others). The studies included did not specify which type of fatigue they measured. In two studies (Jerome et al. 2017; Grospretre et al. 2021) it was described as self-reported ‘fatigue’, in one of these it was classified under ‘physical aspects’ (Grospretre et al. 2021). In the third study it was described as ‘sensations of drowsiness and energy depletion… tiredness’ (Blasche et al. 2018). The definitions used seem to fit well with the definition of mental fatigue so we have clarified this in the results and discussion:
- ‘Self-reported mental fatigue was shown to decrease by 34% [52] to 43% [59]…’
- ‘Students who participated in CMB self-reported improved wellbeing, focus and attention and decreased mental fatigue’.
Round 2
Reviewer 1 Report
The authors have correctly addressed the proposed comments and suggestions.